# Comparison of serum vitamin D level and vitamin D receptor gene *FokI* polymorphism in leprosy patients with and without trophic ulcers: A case-control study

**Paulus Anthony Halim**[1], **Sondang P. Sirait**[1☯*], **Eliza Miranda**[1☯], **Yulia Ariani**[2,3], **Wresti Indriatmi**[1], **Hok Bing Thio**[4]

1 Department of Dermatology and Venereology, Faculty of Medicine Universitas Indonesia, Dr. Cipto Mangunkusumo Hospital, Jakarta, Indonesia, 2 Department of Medical Biology, Faculty of Medicine, Universitas Indonesia, Jakarta, Indonesia, 3 Human Genetic Research Center, Indonesian Medical Education and Research Institute (IMERI), Universitas Indonesia, Jakarta, Indonesia, 4 Department of Dermatology, Erasmus University Medical Center, Rotterdam, The Netherlands

☯ SPS and EM are Joint Senior Authors.
* sondangdr@yahoo.com

## Abstract

### Background

Alterations in the vitamin D axis have been implicated in chronic wounds, but their role in leprosy-related trophic ulcers (TU) is unclear. This study compared serum vitamin D levels and *VDR FokI* (rs2228570) polymorphism between leprosy patients with and without TU, assessed their association with ulcer presence, and examined the correlation between vitamin D levels and ulcer severity.

### Methods

This case-control study involved 82 adult leprosy patients (41 with TU, 41 without) treated at a tertiary referral hospital in Jakarta, Indonesia. Serum 25-hydroxyvitamin D [25(OH)D] levels were measured using chemiluminescence assay, and *FokI* polymorphism was analyzed using polymerase chain reaction-restriction fragment length polymorphism (PCR-RFLP). Ulcer severity was evaluated using the Pressure Ulcer Scale for Healing (PUSH). Associations were assessed through bivariate and multivariate analyses.

### Results

Patients with TU had significantly lower mean serum 25(OH)D levels than those without TU (13.14 vs. 20.18 ng/mL, $p < 0.001$). The homozygous mutant *FokI* genotype (*ff*) was more frequent in the TU group ($p = 0.043$). In multivariable analysis, both low vitamin D status (adjusted odds ratio [aOR] 4.15; 95% confidence interval [CI] 1.41–12.24; $p = 0.010$) and the *FokI* polymorphism (aOR 7.86; 95% CI 1.80–34.36;

**Data availability statement:** All relevant data are within the manuscript and its Supporting Information files.

**Funding:** This work was supported by a grant to SPS from the Directorate of Research and Development, Universitas Indonesia under the Hibah PUTI 2023 scheme (NKB-689/UN2. RST/HKP.05.00/2023). The funder had no role in study design, data collection and analysis, decision to publish, or preparation of the manuscript.

**Competing interests:** The authors have declared that no competing interests exist.

$p = 0.006$) remained independently associated with TU. Vitamin D levels showed a weak inverse correlation with PUSH scores ($r = -0.312$, $p = 0.047$).

## Conclusions

Lower serum vitamin D status and the *FokI* polymorphism were independently associated with TU in leprosy, and a weak inverse correlation was observed between serum vitamin D levels and ulcer severity. While genetic variation is non-modifiable, vitamin D status represents a potentially modifiable factor; hence, these findings suggest that assessment of vitamin D status may be considered as part of comprehensive ulcer care in leprosy.

### Author summary

Leprosy can result in chronic wounds (trophic ulcers) that are difficult to heal. In this study, leprosy patients with ulcers had lower vitamin D levels than those without ulcers, and lower levels were linked to greater ulcer severity. A variant of the vitamin D receptor gene (*FokI* polymorphism), which produces a receptor with reduced transcriptional activity, was also more frequent among patients with ulcers, suggesting a genetic contribution to ulcer susceptibility. While genetic factors are non-modifiable, vitamin D deficiency is potentially correctable. Assessment of vitamin D status may therefore be relevant as part of comprehensive ulcer care in people affected by leprosy.

## Introduction

Leprosy, or Hansen's disease (HD), is a chronic infection caused by *Mycobacterium leprae* and *M. lepromatosis*, that mainly affects the skin and peripheral nerves. Although global control efforts have reduced incidence, the disease remains endemic in countries like Brazil, India, and Indonesia, that together accounts for approximately 80% of new cases worldwide [1]. Vulnerable populations, particularly refugee and immigrant groups, also carry a disproportionately high leprosy burden. A recent work reported that 25.9% of all identified dermatological conditions in these populations were attributable to leprosy [2]. In 2023, Indonesia reported 14,376 new cases, with 3.0 new grade 2 disability (G2D) cases per million population, far exceeding the World Health Organization (WHO) 2030 target of 0.12 [1]. One of the most common cause of disability among leprosy patients is trophic ulcer (TU), affecting 10–20% of patients [3,4]. These chronic wounds, often on the plantar surface, result from neuropathy, foot deformities, impaired sensation, and comorbidities [5]. They are notoriously difficult to manage and are associated with frequent recurrence, infection, and even malignant transformation [6].

Vitamin D, a secosteroid hormone, has demonstrated a key role in immune modulation and wound healing [7,8]. In damaged tissue, the circulating form, 25-hydroxyvitamin

D [25(OH)D], is converted to calcitriol, binds to the vitamin D receptor (VDR), and induces antimicrobial peptides and cytokines involved in tissue repair [9–11]. Vitamin D deficiency has been linked to poor healing in chronic wounds (e.g., diabetic and venous ulcers) [12–14], with some studies showing improvement following supplementation [15,16].

Despite abundant sunlight exposure, vitamin D deficiency is prevalent in the general population in Indonesia and other Southeast Asian countries, with reported rates ranging from 6% to 70% [17–19]. In addition, a study from our center reported a 71.4% prevalence of vitamin D deficiency among newly diagnosed leprosy patients [20]. Furthermore, a previous study conducted in Medan, Indonesia demonstrated lower vitamin D levels among leprosy patients compared to healthy controls [21]. However, whether vitamin D status differs specifically in leprosy patients with TU remains unclear.

Various genetic polymorphisms of the *VDR* gene may influence receptor function and downstream effects of vitamin D. Among these, the *FokI* polymorphism (rs2228570) is unique in that it alters the translation initiation site, resulting in structural variation of the receptor [22,23]. The *ff* genotype produces a longer, less active receptor, and has been associated with leprosy susceptibility and chronic ulcers in other conditions [24–26].

While previous studies have linked vitamin D deficiency and *VDR* polymorphisms to diabetic foot ulcers [25,27] and leprosy risk [21,28–30], their role in leprosy-related TU remains unexplored. This study aims to assess differences in serum 25(OH)D levels and the distribution of the *FokI* polymorphism between leprosy patients with and without TU, assess their association with TU incidence, and determine the correlation between vitamin D levels and ulcer severity.

## Materials and methods

### Ethics statement

The study protocol adhered to the Declaration of Helsinki and was approved by the Health Research Ethics Committee of the Faculty of Medicine, Universitas Indonesia – Dr. Cipto Mangunkusumo Hospital (No. KET-82/UN2.F1/ETIK/PPM.00.02/2024). All participants provided written informed consent following an explanation of the study objectives, procedures, and risks.

### Study design

The full study protocol has been previously published [31]. In brief, this single-center, observational, analytic case-control study was conducted at the Dermatology and Venereology Clinic of Dr. Cipto Mangunkusumo Hospital, Jakarta, Indonesia. Adult leprosy patients, with and without TU, were consecutively recruited.

### Participants

Eligible participants were adults aged 18 years or older who met the World Health Organization (WHO) diagnostic criteria for leprosy, including newly diagnosed individuals, those currently receiving treatment, or those released from treatment. The case group included patients with TU meeting all of the following criteria: ulcer duration of at least four weeks without a history of recent sharp debridement or callus thinning; ulcer depth not exceeding the subcutaneous layer; and absence of secondary infection or only mild signs of infection (i.e., surrounding erythema <2 cm, minimal edema or purulent discharge, and no warmth on palpation). Exclusion criteria were: recent consumption of vitamin D supplements within the past four weeks; current signs or symptoms of leprosy reactions; or comorbidities that may confound ulcer healing or vitamin D metabolism, including systemic infections, diabetes mellitus, severe hepatic dysfunction, end-stage renal disease, malignancy, or autoimmune disorders.

### Data collection

Structured interviews and physical examinations were conducted using standardized forms. Collected data included sociodemographic details, leprosy and ulcer history, duration of sun exposure, and smoking status. Detailed dietary

vitamin D intake was not assessed, as recall-based dietary reporting in this population can be unreliable. Moreover, sun exposure represents the predominant source of circulating vitamin D [32]. Physical examination included anthropometric measurements, ulcer characteristics, presence of deformities, and sensory testing using 0.2 g (blue) and 2 g (purple) Semmes-Weinstein monofilaments. Ulcer photographs were taken using an iPhone 12 camera under standardized conditions. Ulcer area was calculated by multiplying two perpendicular diameters measured with a digital caliper, with the largest ulcer per subject selected for analysis. Ulcer depth and severity were assessed using the Pressure Ulcer Scoring System (PUSS) and the Pressure Ulcer Scale for Healing (PUSH) tool, respectively [33].

## Sample collection and laboratory procedures

Venous blood (6 mL) was collected in ethylenediaminetetraacetic acid (EDTA) and plain tubes, stored at 2–8°C, and processed within 2 hours. Serum 25(OH)D concentrations were determined from plain tube samples using a chemiluminescent immunoassay (Architect 25(OH)D kit; Abbott Diagnostics, IL, USA). The classification of vitamin D status was based on Endocrine Society guidelines (<20 ng/mL for deficiency, 20–30 ng/mL for insufficiency, >30 ng/mL for sufficiency) [34], as locally validated cutoffs are unavailable. Severe deficiency was defined as <10 ng/mL in line with recent evidence [35].

The full procedure for *FokI* genotyping using the PCR–restriction fragment length polymorphism (PCR-RFLP) method is detailed in the published protocol [31]. In short, genomic DNA was extracted from EDTA blood using the salting-out method, followed by PCR with primers described by Soroush et al. [19]. The amplified products were digested with the *FokI* enzyme (New England Biolabs, MA, USA) and visualized on 2% agarose gel. Genotypes were determined based on band patterns (S1 Fig): 192 and 58 bp for the homozygous mutant genotype (*ff*), 250, 192, and 58 bp for the heterozygous genotype (*Ff*), and a single 250 bp fragment for the homozygous wild-type genotype (*FF*).

## Sample size estimates

Sample size was calculated to detect differences in serum vitamin D levels, the frequency of *FokI* polymorphism, and correlations with ulcer severity, using standard formulas with 80% power and 5% significance. Based on prior data, a minimum of 41 subjects per group was required, yielding a total of 82 participants [31].

## Statistical analysis

Data were analyzed using STATA v16.0 (StataCorp LLC, College Station, TX, USA) and SNPStats (Catalan Institute of Oncology, Barcelona, Spain). Bivariate and multivariate analyses were conducted and significance was considered at $p < 0.05$. Variables with $p < 0.25$ in bivariate analysis were deemed eligible for inclusion in the multivariate logistic regression model, following a purposeful selection approach. All selected variables were entered into the final model to allow adjustment for potential confounding. Because this study evaluated a single pre-specified candidate polymorphism (*FokI*), formal adjustment for multiple testing was not applied. Inheritance model analyses were performed in an exploratory manner.

## Results

### Sociodemographic and clinical characteristics

A total of 82 participants were recruited between March and July 2024, during the Indonesian dry season, comprising 41 leprosy patients with TU and 41 without. The selection process is summarized in the study flow diagram (Fig 1). Sociodemographic characteristics are presented in Table 1. No statistically significant differences were observed between the groups in terms of sex, age, occupation, marital status, smoking history, or duration of sun exposure. However, patients with TU had significantly lower education levels ($p < 0.001$) and a higher proportion of household incomes below the regional minimum wage ($p = 0.010$).

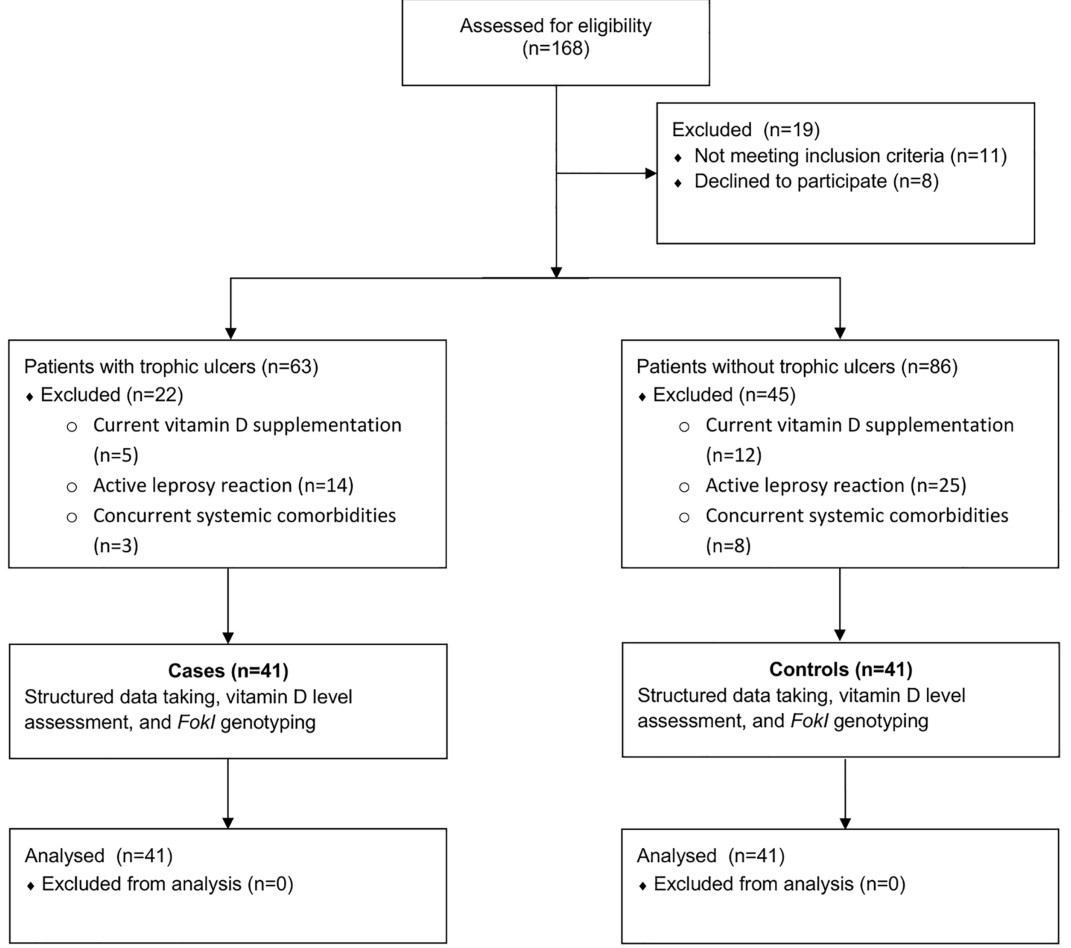

**Fig 1. Study flow diagram.**

The clinical characteristics of both groups are detailed in Table 2. Subjects in the TU group had significantly longer disease duration (median 10 vs. 2.25 years, $p<0.001$), higher prevalence of sensory neuropathy in the hands (97.6% vs. 51.2%, $p<0.001$) and feet (100% vs. 78%, $p=0.002$), and higher prevalence of non-ulcer deformities (70.7% vs. 12.2%, $p<0.001$). No significant differences were observed in nutritional status, disease classification (WHO and Ridley-Jopling), treatment status, history of leprosy reactions, and hypertension

### Ulcer characteristics

There were 69 ulcers among the 41 subjects with TU. For consistency in analysis, only the largest ulcer per patient was evaluated, with aggregated data presented in Table 3. The median ulcer area was 1.89 cm², and the majority (78.1%) extended to the subcutaneous tissue. Most ulcers were located on the lower extremities (90.2%), particularly on the fore-foot (41.5%). The median ulcer duration was four months, and the median PUSH score was 7 (range 3–14). The majority of ulcers exhibited mild to moderate exudate (70.7%) and had a granulation tissue base (73.2%). Callus formation sur-rounding the ulcer was observed in 82.9% of cases.

**Table 1. Sociodemographic characteristics of study subjects.**

| Characteristics | Leprosy with TU (n=41) | Leprosy without TU (n=41) | p-value |
|---|---|---|---|
| Sex, n (%) | | | 0.822[a] |
| Male | 24 (58.5) | 25 (61.0) | |
| Female | 17 (41.5) | 16 (39.0) | |
| Age, years (mean±SD) | 43.83±11.38 | 40.83±13.52 | 0.280[b] |
| Ethnic group, n (%) | | | 0.218[a] |
| Javanese | 15 (36.6) | 14 (34.1) | |
| Betawi | 10 (24.4) | 9 (22.0) | |
| Sundanese | 11 (26.8) | 6 (14.6) | |
| Others | 5 (12.2) | 12 (29.3) | |
| Education level, n (%) | | | **<0.001[a]** |
| None or primary | 18 (43.9) | 3 (7.3) | |
| Secondary | 20 (48.8) | 23 (56.1) | |
| Tertiary | 3 (7.3) | 15 (36.6) | |
| Occupation, n (%) | | | 0.334[a] |
| Unemployed | 18 (43.9) | 14 (34.1) | |
| Employee | 18 (43.9) | 17 (41.5) | |
| Self-employed | 5 (12.2) | 10 (24.4) | |
| Marital status, n (%) | | | 0.414[a] |
| Unmarried | 7 (17.1) | 10 (24.4) | |
| Married | 34 (82.9) | 31 (75.6) | |
| Household income, n (%) | | | **0.010[a]** |
| <Jakarta RMW | 33 (80.5) | 22 (53.7) | |
| ≥ Jakarta RMW | 8 (19.5) | 19 (46.3) | |
| Smoking history, n (%) | | | 0.822[a] |
| No | 24 (58.5) | 25 (61.0) | |
| Yes | 17 (41.5) | 16 (39.0) | |
| Sun exposure duration, hours/week [median (range)] | 5.0 (0.5–35.0) | 5.5 (1.0–42.0) | 0.613[c] |

RMW, 2024 Jakarta monthly regional minimum wage (Rp 5,067,381 or circa US$ 301); SD, standard deviation; TU, trophic ulcer; [a]Chi-square test; [b]independent t-test; [c]Mann-Whitney U test.

## Serum 25(OH)D levels

The mean serum 25(OH)D concentration was notably lower in individuals with trophic ulcers (TU) compared to those without (13.14±5.88 ng/mL vs. 20.18±7.19 ng/mL, $p<0.001$), as represented in Fig 2. Furthermore, vitamin D deficiency or severe deficiency was significantly more prevalent among patients in the TU group (82.9% vs. 36.6%, $p<0.001$), as shown in Table 4. Exploratory subgroup analyses did not demonstrate significant differences in serum 25(OH)D levels between BT/BB and BL/LL subtypes in the overall cohort or within TU and non-TU groups.

## *FokI* polymorphism

Genotype and allele frequencies of the *FokI* polymorphism are presented in Table 4. Genotype distributions in both the case ($p=0.76$) and control ($p=0.18$) groups conformed to Hardy–Weinberg equilibrium. The homozygous mutant genotype (*ff*) was nearly four times more prevalent in patients with TU than in those without (26.8% vs. 7.3%, $p=0.043$). In contrast, the wild-type genotype (*FF*) was more frequently observed in the non-TU group (34.2% vs. 19.5%). The mutant allele (*f*) was also more frequently observed in the TU group than in the control group (53.7% vs. 36.6%, $p=0.028$).

**Table 2. Clinical characteristics of study subjects.**

| Characteristics | Leprosy with TU (n=41) | Leprosy without TU (n=41) | p-value |
|---|---|---|---|
| Body mass index, kg/m² (mean±SD) | 21.94±3.67 | 23.19±3.94 | 0.140[a] |
| Nutritional status, n (%) | | | 0.284[b] |
| Underweight | 7 (17.1) | 3 (7.3) | |
| Normal weight | 20 (48.8) | 17 (41.5) | |
| Overweight | 3 (7.3) | 7 (17.1) | |
| Obesity | 11 (26.8) | 14 (34.1) | |
| Duration of leprosy, n (%) | | | **<0.001[b]** |
| ≤ 1 year | 2 (4.9) | 11 (26.8) | |
| 1–5 year | 14 (34.1) | 22 (53.7) | |
| >5 year | 15 (61.0) | 8 (19.5) | |
| WHO classification, n (%) | | | 1.000[c] |
| Paucibacillary | 0 (0) | 1 (2.4) | |
| Multibacillary | 41 (100) | 40 (97.6) | |
| Ridley-Jopling classification, n (%) | | | 1.000[c] |
| BT | 16 (39.0) | 15 (36.6) | |
| BB | 2 (4.9) | 3 (7.3) | |
| BL | 13 (31.7) | 14 (34.1) | |
| LL | 10 (24.4) | 9 (22.0) | |
| Treatment status, n (%) | | | 0.077[b] |
| Under treatment | 16 (39.0) | 24 (58.5) | |
| Released from treatment | 25 (61.0) | 17 (41.5) | |
| History of reaction, n (%) | | | 0.201[c] |
| No | 5 (12.2) | 6 (14.6) | |
| Type 1 | 21 (51.2) | 28 (68.3) | |
| Type 2 | 14 (34.2) | 6 (14.6) | |
| Both | 1 (2.4) | 1 (2.4) | |
| Hand sensory neuropathy, n (%) | | | **<0.001[c]** |
| No | 1 (2.4) | 20 (48.8) | |
| Yes | 40 (97.6) | 21 (51.2) | |
| Foot sensory neuropathy, n (%) | | | **0.002[c]** |
| No | 0 (0) | 9 (22) | |
| Yes | 41 (100) | 32 (78) | |
| Non-ulcer deformity, n (%) | | | **<0.001[b]** |
| No | 12 (29.3) | 36 (87.8) | |
| Yes | 29 (70.7) | 5 (12.2) | |
| Hypertension, n (%) | | | 0.135[b] |
| No | 27 (65.9) | 33 (80.5) | |
| Yes | 14 (34.1) | 8 (19.5) | |

TU, trophic ulcer; SD, standard deviation; [a]Independent t-test; [b]Chi-square test; [c]Fisher's exact test

## Association between vitamin D and *FokI* polymorphism with TU

In bivariate analysis, vitamin D status, *FokI* polymorphism, educational level, household income, leprosy duration, and of non-ulcer deformity were associated with TU (Table 5). These variables, together with nutritional status, were included in

**Table 3. Characteristics of the largest ulcer among the trophic ulcer group (n=41).**

| Characteristics | Value |
|---|---|
| Vertical diameter, cm [median (range)] | 1.43 (0.20–9.37) |
| Horizontal diameter, cm [median (range)] | 1.42 (0.20–7.05) |
| Area, cm$^2$ [median (range)] | 1.89 (0.04–41.23) |
| Depth, n (%) | |
| Grade 2 – dermis | 9 (21.9) |
| Grade 3 – subcutaneous | 32 (78.1) |
| Ulcer duration, n (%) | |
| 1–6 months | 25 (61.0) |
| 6 months– 2 years | 11 (26.8) |
| 2–5 years | 4 (9.8) |
| >5 years | 1 (2.4) |
| Location, n (%) | |
| Forefoot | 17 (41.5) |
| Ankle | 5 (12.2) |
| Hallux | 5 (12.2) |
| Midfoot | 3 (7.3) |
| Toes | 2 (4.9) |
| Hindfoot | 2 (4.9) |
| Dorsum of foot | 1 (2.4) |
| Palm | 2 (4.9) |
| Fingers | 2 (4.9) |
| Others | 2 (4.9) |
| Total PUSH score [median (range)] | 7 (3–14) |
| Exudate subscore, n (%) | |
| 0 – None | 11 (26.8) |
| 1 – Mild | 16 (39.0) |
| 2 – Moderate | 13 (31.7) |
| 3 – Severe | 1 (2.4) |
| Wound bed subscore, n (%) | |
| 1 – Epithelial tissue | 3 (7.3) |
| 2 – Granulation tissue | 30 (73.2) |
| 3 – Slough | 7 (17.1) |
| 4 – Necrotic tissue | 1 (2.4) |
| Peri-ulcer callus formation, n (%) | |
| No | 7 (17.1) |
| Yes | 34 (82.9) |

the multivariable logistic regression to control for potential confounding. After adjustment, low vitamin D status remained independently associated with TU (adjusted odds ratio [aOR] 4.15; 95% CI: 1.41–12.24; $p=0.010$), as did the presence of the *FokI* polymorphism (aOR 7.86; 95% CI: 1.80–34.36; $p=0.006$). Educational level showed a protective association with TU, while non-ulcer deformity remained positively associated. Leprosy duration did not remain statistically significant after adjustment but was retained in the model due to its clinical relevance as a potential confounder.

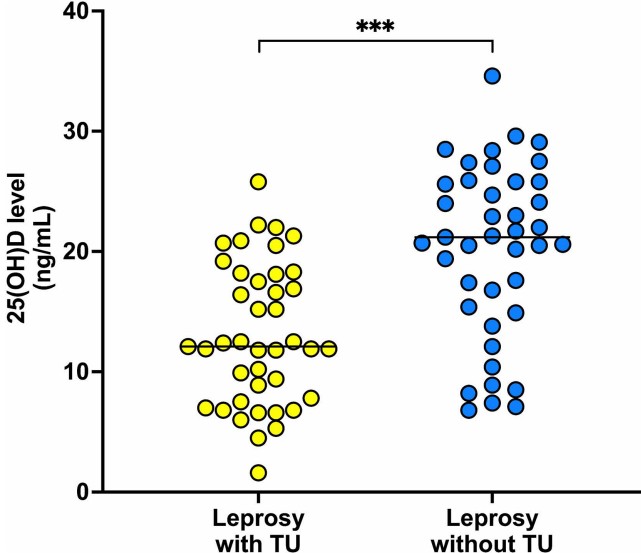

**Fig 2. Scatter plot showing serum 25(OH)D levels in leprosy patients with and without trophic ulcer (***, *p*<0.001).**

**Table 4. Vitamin D status and *FokI* polymorphism genotype and allele frequencies in leprosy patients with and without trophic ulcers.**

|  | Leprosy with TU (n=41) | Leprosy without TU (n=41) | p-value |
|---|---|---|---|
| Vitamin D status, n (%) |  |  | **<0.001ᵃ** |
| Sufficiency | 0 (0.0) | 1 (2.4) |  |
| Insufficiency | 7 (17.1) | 25 (61.0) |  |
| Deficiency | 20 (48.8) | 9 (22.0) |  |
| Severe deficiency | 14 (34.1) | 6 (14.6) |  |
| *FokI* genotype, n (%) |  |  | **0.043ᵇ** |
| *FF* | 8 (19.5) | 14 (34.2) |  |
| *Ff* | 22 (53.7) | 24 (58.5) |  |
| *ff* | 11 (26.8) | 3 (7.3) |  |
| Allele, n (%) |  |  | **0.028ᵇ** |
| *F* | 38 (46.3) | 52 (63.4) |  |
| *f* | 44 (53.7) | 30 (36.6) |  |

TU, trophic ulcer; ᵃFisher's exact test, ᵇChi-square test

### Correlation between vitamin D levels and ulcer severity

A weak but statistically significant inverse correlation was found between serum 25(OH)D levels and PUSH scores among leprosy patients with TU (Pearson's $r=-0.312$, $p=0.047$), as shown in Fig 3A. Exploratory subgroup analyses (Fig 3B, 3C) demonstrated a moderate inverse correlation in BT/BB subjects (n=18; $r=-0.605$, $p=0.008$), whereas no significant correlation was observed in BL/LL subjects (n=23; $r=-0.079$, $p=0.719$).

### Genetic model analysis

Different inheritance models of the *FokI* polymorphism were evaluated for their association with TU (Table 6). Statistically significant associations were observed under the codominant ($p=0.037$), recessive ($p=0.016$), and log-additive ($p=0.017$)

**Table 5. Multivariate analyses of variables associated with trophic ulcers.**

| Variable | Bivariate analysis | | Multivariate analysis | |
|---|---|---|---|---|
| | Crude OR (95% CI) | p-value | Adjusted OR (95% CI) | p-value |
| Vitamin D status | 3.26 (1.71–6.20) | **<0.001** | 4.15 (1.41-12.24) | **0.010** |
| *FokI* polymorphism | 2.32 (1.13–4.76) | **<0.001** | 7.86 (1.80-34.36) | **0.006** |
| Education | 0.19 (0.08–0.44) | **<0.001** | 0.19 (0.04–0.92) | **0.039** |
| Household income | 0.28 (0.10-0.75) | **0.012** | 0.49 (0.06-3.93) | 0.504 |
| Nutritional status | 0.73 (0.48–1.11) | 0.137 | 1.05 (0.50–2.24) | 0.893 |
| Leprosy duration | 1.15 (1.05-1.26) | **0.004** | 1.13 (0.97-1.31) | 0.119 |
| Presence of non-ulcer deformity | 17.40 (5.50-55.07) | **<0.001** | 6.72 (1.10-41.18) | **0.039** |

OR, odds ratio; 95% CI, 95% confidence interval

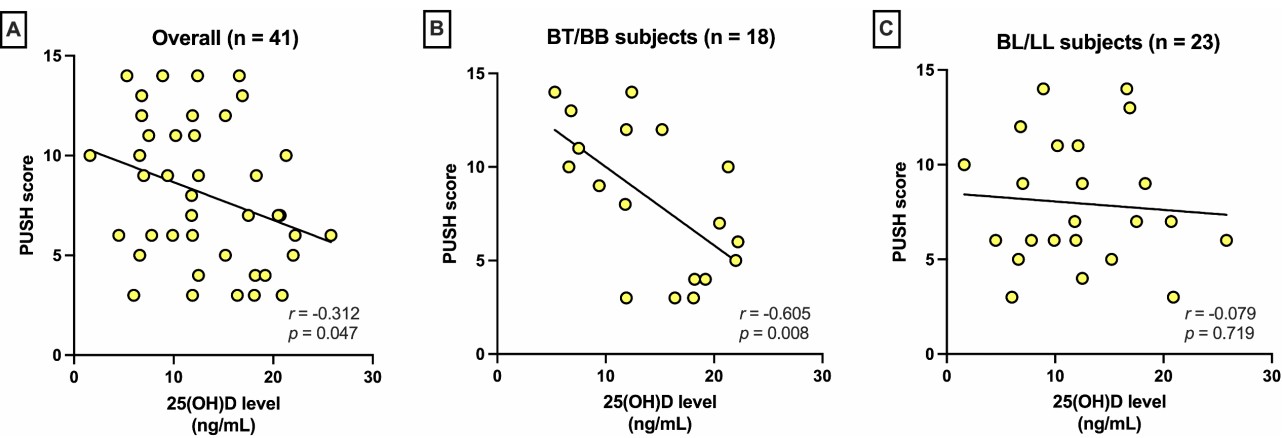

**Fig 3. Correlation between serum 25(OH)D levels and trophic ulcer severity, as assessed using the Pressure Ulcer Scale for Healing (PUSH) scoring.** (A) Overall, a weak inverse correlation was observed. In exploratory subgroup analyses, a stronger inverse correlation was found in BT/BB subjects (B), whereas no correlation was observed in BL/LL subjects (C).

models. The recessive model showed the best fit based on the lowest Akaike and Bayesian Information Criterion values. In this model, patients with the homozygous mutant genotype (*ff*) had a 4.64-fold higher risk of developing TU compared to those with *FF* or *Ff* genotypes.

## Discussion

This study highlights an observed association between vitamin D status and receptor polymorphism with TU development among leprosy patients. Serum 25(OH)D levels were lower and the *FokI* polymorphism was more frequent in patients with TU, and both remained associated with TU after adjustment for confounders. A weak inverse correlation was also

**Table 6. Genetic models of *FokI* polymorphism associated with trophic ulcer among leprosy patients.**

| Model | Genotype | Leprosy with TU (n=41) | Leprosy without TU (n=41) | Adjusted OR (IK 95%) | p-value | AIC | BIC |
|---|---|---|---|---|---|---|---|
| Codominant | *FF* | 8 (19.5%) | 14 (34.1%) | Reference | **0.037** | 113.1 | 120.3 |
| | *Ff* | 22 (53.7%) | 24 (58.5%) | 1.60 (0.56–4.56) | | | |
| | *ff* | 11 (26.8%) | 3 (7.3%) | 6.42 (1.37–30.06) | | | |
| Dominant | *FF* | 8 (19.5%) | 14 (34.1%) | Reference | 0.130 | 115.4 | 120.2 |
| | *Ff – ff* | 33 (80.5%) | 27 (65.8%) | 2.14 (0.78–5.85) | | | |
| Recessive | *FF – Ff* | 30 (73.2%) | 38 (92.7%) | Reference | **0.016** | 111.9 | 116.7 |
| | *ff* | 11 (26.8%) | 3 (7.3%) | 4.64 (1.19–18.16) | | | |
| Overdominant | *FF – ff* | 19 (46.3%) | 17 (41.5%) | Reference | 0.660 | 117.5 | 122.3 |
| | *Ff* | 22 (53.7%) | 24 (58.5%) | 0.82 (0.34–1.96) | | | |
| Log-additive | – | – | – | 2.32 (1.13–4.76) | **0.017** | 112 | 116.8 |

TU, trophic ulcer; OR, odds ratio; AIC, Akaike Information Criterion; BIC, Bayesian Information Criterion.

observed between vitamin D levels and ulcer severity. However, causality cannot be inferred due to the case-control design.

Despite the absence of sample matching, comparable age and sex distributions between groups likely minimized confounding. Some sociodemographic disparities persisted, with significantly more patients in the TU group having lower educational attainment and household incomes below the regional minimum wage. These findings align with prior evidence that low socioeconomic status impairs access to healthcare and delays ulcer prevention and management [5]. Clinically, patients with TU had a significantly longer disease duration and higher prevalence of sensory neuropathy and deformities. These features are well-established risk factors for ulcer formation due to repeated unnoticed trauma and altered biomechanical pressure points [3,5,36], particularly on weight-bearing areas like the forefoot, which was the most common ulcer site in this study. The median ulcer duration of four months, with a substantial portion persisting for several years, underscores the chronic and refractory nature of TU in leprosy.

Patients with TU exhibited significantly lower serum 25(OH)D levels, supporting our primary hypothesis. Vitamin D status also remained independently associated with TU after adjustment for potential confounders. To our knowledge, this is the first investigation to explore this association specifically in individuals with leprosy. As this was a case-control study, temporal direction cannot be determined, and the findings should not be interpreted as evidence of causality. Low vitamin D levels have similarly been associated with impaired healing in other chronic ulcers, including diabetic, venous, and pressure ulcers, suggesting a broader role of vitamin D in wound biology [12–14,37,38]. However, given the multifactorial nature of these conditions, vitamin D is more appropriately regarded as a contributory factor rather than a sole causal determinant.

In our study, lower vitamin D levels were weakly correlated with increased TU severity, consistent with findings in diabetic foot ulcers (DFU) [12,39,40]. Although the observed correlation was weak and of uncertain clinical significance, it may reflect a potential contribution of vitamin D status to ulcer progression. In exploratory subgroup analyses, the inverse correlation between vitamin D levels and ulcer severity appeared stronger among BT/BB subjects but was not observed in BL/LL patients, suggesting greater clinical relevance in the former group. One possible explanation for this observation is that tuberculoid leprosy forms are immunologically characterized by relatively preserved cell-mediated immunity, whereas lepromatous disease is associated with impaired cellular responses, in which vitamin D-mediated effect might be less clinically apparent [28]. In addition, ulcer severity in BL/LL patients may be more strongly influenced by greater disease

burden, including nerve and structural damage, thereby limiting the contribution of vitamin D. However, given the modest subgroup size and post-hoc nature of this analysis, these findings should be considered hypothesis-generating.

Vitamin D, through its active form that binds to the cytoplasmic vitamin D receptor, regulates genes involved in antimicrobial peptide production, inflammatory response, and tissue repair. Various vitamin D-dependent pathways are engaged throughout these processes, thus disturbances caused by low vitamin D levels or receptor polymorphisms can interfere with these functions. In leprosy, two Brazilian studies have shown that both vitamin D deficiency and *VDR* polymorphisms are associated with increased disease susceptibility [41,42]. Evidence from chronic ulcer research also indicates that vitamin D contributes to wound healing through effects on innate immunity, cytokine regulation, angiogenesis, and keratinocyte proliferation [7,9,10]. Vitamin D supplementation has been linked to improved outcomes in chronic ulcers [15,43], and agents acting downstream of vitamin D have shown clinical benefit. In a randomized trial, Miranda et al. [44] reported that topical LL-37, a peptide induced by vitamin D signaling, accelerated wound granulation in DFU.

Furthermore, vitamin D influences several pathways that may contribute to TU formation in leprosy. Reduced vitamin D activity weakens vitamin D-dependent innate immune pathways, lowering LL-37 and other antimicrobial peptides [11,45]. This impaired response may compromise cutaneous host defence and contribute to the microbiome dysbiosis and metabolic disturbances reported in TU [46]. Adequate vitamin D signalling also supports TLR2/1-mediated macrophage killing of *M. leprae* [47] and promotes nerve repair through Schwann cells [48]; therefore, deficiency may exacerbate neural damage. As neuropathy progresses, sensory loss permits repeated unrecognised trauma, while autonomic fiber involvement can reduce local perfusion and sweating, weakening the epidermal barrier [49]. These fibers also release neuropeptides that promote vasodilation, chemotaxis, and keratinocyte repair, and loss of these signals has been shown to delay wound healing in diabetic neuropathy [50]. While a similar neuropeptide-related disturbance is plausible in leprosy, it remains understudied.

Although seasonal variation in vitamin D levels is well recognized in higher-latitude countries, its impact in this study is likely minimal. Indonesia, an equatorial country, receives relatively constant daylight duration and high UV indices throughout the year, including in Jakarta where seasonal fluctuations are modest [51]. Studies from Indonesia and other Southeast Asian countries have not demonstrated meaningful seasonal differences in serum 25(OH)D levels [52,53]. In this study, all participants were recruited during the dry season, reducing intra-study variation in UV exposure. While the wet season may theoretically reduce sun exposure, these factors indicate that seasonal bias is unlikely to have significantly affected our findings.

The homozygous mutant (*ff*) genotype and allele (*f*) of the *FokI* polymorphism more frequent among TU patients. Although the *FokI* polymorphism remained associated with TU after multivariable adjustment, the association appeared less robust than that of vitamin D status. suggesting that its clinical relevance may be modulated by modifiable factors such as vitamin D levels. Similar patterns have been reported in DFU, where associations observed in unadjusted analyses were attenuated after accounting for vitamin D status and other covariates [25].

Mechanistically, the *ff* genotype encodes a longer VDR isoform with reduced transcriptional activity [22]. Under the recessive model, individuals with the *Ff* genotype still produce the shorter, more active isoform, which may explain the limited biological effect of *FokI* heterozygosity. Reduced VDR activity in *ff* homozygotes could diminish vitamin D-dependent antimicrobial, immunomodulatory, and tissue repair pathways [7,9], contributing to neural damage or delayed wound healing in leprosy. Still, these effects may be insufficient to influence ulcer development without concurrent vitamin D deficiency. Finally, TU development is multifactorial. While vitamin D status represents one potentially modifiable factor, educational level likely reflects broader social determinants of health. Non-ulcer deformities may be partially mitigated through strengthened prevention of disability (POD) care, although established deformities are largely irreversible.

While the recessive model showed the strongest association in our analysis, genetic model testing was exploratory and should therefore be interpreted cautiously. The additional significance observed under the log-additive model may suggest a potential incremental effect of the mutant allele and could be consistent with a broader polygenic mechanism on ulcer

susceptibility. These findings require confirmation in larger cohorts. Future genome-wide association studies (GWAS) are warranted to identify additional genetic loci involved in immune regulation, nerve damage, and wound repair as potential risk factors for TU in leprosy patients.

### Strengths and limitations

This study is the first to investigate both serum vitamin D levels and *FokI* polymorphism in leprosy patients with and without TU. It also represents an early effort to explore the potential involvement of the vitamin D axis in TU development within this population. Nonetheless, several limitations should be acknowledged. The single-center design may limit generalizability, and the case-control methodology precludes determination of temporal or causal relationships. Although recalled weekly sun exposure duration was similar between groups, reverse causality cannot be excluded, as chronic ulcers may restrict outdoor mobility and reduce effective sun exposure over time, potentially resulting in lower vitamin D levels. In addition, chronic inflammatory processes associated with leprosy and persistent ulcers may influence vitamin D metabolism. Several baseline characteristics differed between groups; however, clinically relevant variables were addressed through multivariable adjustment. While the sample size was determined a priori based on the primary objectives, a larger cohort may improve the precision of effect estimates, particularly for genetic models. Finally, only one *VDR* polymorphism was examined, while other potentially relevant *VDR* and non-*VDR* variants were not assessed and warrant future study.

### Conclusion

This study demonstrates that lower serum vitamin D status and the *FokI* polymorphism were independently associated with the presence of TU in patients with leprosy. Furthermore, a weak inverse correlation was observed between serum vitamin D levels and ulcer severity. While genetic variation may contribute to ulcer susceptibility, vitamin D status represents a potentially modifiable parameter in clinical practice. These findings support consideration of vitamin D status assessment within comprehensive ulcer care in leprosy. However, prospective cohort studies and interventional trials are required to clarify temporal relationships and to determine whether correction of vitamin D deficiency can reduce ulcer incidence or severity in this population.

### Supporting information

**S1 Fig. Electrophoresis visualisation of *FokI* restriction enzyme analysis. bp, base pair; M, GeneRuler DNA ladder.** (PDF)

**S1 File. STROBEChecklist.** Strengthening the Reporting of Observational Studies in Epidemiology (STROBE) checklist for the manuscript. (PDF)

**S1 Data. Figshare link for the study data set.** (DOCX)

### Author contributions

**Conceptualization:** Paulus Anthony Halim, Sondang P. Sirait, Eliza Miranda, Yulia Ariani.

**Data curation:** Paulus Anthony Halim, Wresti Indriatmi.

**Formal analysis:** Paulus Anthony Halim, Yulia Ariani, Wresti Indriatmi.

**Funding acquisition:** Paulus Anthony Halim, Sondang P. Sirait, Eliza Miranda, Hok Bing Thio.

**Investigation:** Paulus Anthony Halim.

**Methodology:** Paulus Anthony Halim, Sondang P. Sirait, Eliza Miranda, Yulia Ariani, Wresti Indriatmi, Hok Bing Thio.

**Project administration:** Paulus Anthony Halim.

**Resources:** Paulus Anthony Halim, Sondang P. Sirait, Eliza Miranda, Yulia Ariani.

**Supervision:** Sondang P. Sirait, Eliza Miranda, Yulia Ariani, Hok Bing Thio.

**Validation:** Sondang P. Sirait, Eliza Miranda, Yulia Ariani, Wresti Indriatmi.

**Visualization:** Paulus Anthony Halim.

**Writing – original draft:** Paulus Anthony Halim.

**Writing – review & editing:** Paulus Anthony Halim, Sondang P. Sirait, Eliza Miranda, Yulia Ariani, Wresti Indriatmi, Hok Bing Thio.

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
