## [Decision Letter · Decision Letter 0]

14 Feb 2026

Response to Reviewers'. This file does not need to include responses to any formatting updates and technical items listed in the 'Journal Requirements' section below.'. This file does not need to include responses to any formatting updates and technical items listed in the 'Journal Requirements' section below. * A marked-up copy of your manuscript that highlights changes made to the original version. You should upload this as a separate file labeled 'Revised Manuscript with Track Changes'.'. * An unmarked version of your revised paper without tracked changes. You should upload this as a separate file labeled 'Manuscript'.'. If you would like to make changes to your financial disclosure, competing interests statement, or data availability statement, please make these updates within the submission form at the time of resubmission. Guidelines for resubmitting your figure files are available below the reviewer comments at the end of this letter. We look forward to receiving your revised manuscript. Kind regards, Mohammad Jokar, DVMGuest EditorPLOS Neglected Tropical Diseases Stuart BlacksellSection EditorPLOS Neglected Tropical Diseases

Shaden Kamhawi

co-Editor-in-Chief

Paul Brindley

co-Editor-in-Chief

**Journal Requirements:**

At this stage, the following Authors/Authors require contributions: Paulus Anthony Halim, Sondang P. Sirait, Eliza Miranda, Yulia Ariani, Wresti Indriatmi, and Hok Bing Thio. Please ensure that the full contributions of each author are acknowledged in the "Add/Edit/Remove Authors" section of our submission form.

4) Please amend your detailed Financial Disclosure statement. This is published with the article. It must therefore be completed in full sentences and contain the exact wording you wish to be published.

State the initials, alongside each funding source, of each author to receive each grant. For example: "This work was supported by the National Institutes of Health (####### to AM; ###### to CJ) and the National Science Foundation (###### to AM).".

**Reviewers' comments:** Reviewer's Responses to Questions

**Key Review Criteria Required for Acceptance?**

**Methods**

-Are the objectives of the study clearly articulated with a clear testable hypothesis stated?

-Is the study design appropriate to address the stated objectives?

-Is the population clearly described and appropriate for the hypothesis being tested?

-Is the sample size sufficient to ensure adequate power to address the hypothesis being tested?

-Were correct statistical analysis used to support conclusions?

-Are there concerns about ethical or regulatory requirements being met?

Reviewer #1: The purpose of this study is clearly explained.

The research design used in this study aligns with the objectives to be achieved.

The proposed hypothesis is clear and in accordance with the hypothesis proposed by the researcher.

The sample size for a case-control design between patients with and without the disease is appropriate and sufficient to ensure power and answer the hypothesis being tested.

The statistical analysis used reflects the overall conclusions.

There are no ethical issues in this research and it complies with applicable regulations.

Reviewer #2: All questions are addressed in full

Reviewer #3: The objectives and hypothesis are stated clearly. Power analysis 80% power and 5% significance performed and sufficient for the study. Approved by the Ethics Committee of Universitas Indonesia (No. KET-82/UN2.F1/ETIK/PPM.00.02/2024).

No major new analyses required.

Reviewer #4: The objective and study design are well explained, and the manuscript is written in a standard format with details.

The only concern I have is the prevalence of Vit D level in general population in Indonesia. I request the researchers to add the prevalence of Vit D level in non-leprosy population and then compare with leprosy population. It seems that Vit D level is low in general population in most of the south Asian countries.

Reviewer #5: - The sample size appears adequate to detect differences in serum vitamin D levels between groups. However, it may be insufficient for robust genetic association analyses, as indicated by wide confidence intervals and low genotype frequencies. The absence of a formal power calculation for the genetic component limits inference, and these findings should be interpreted as exploratory.

- Several genetic comparisons were conducted, including genotype, allele, and inheritance model analyses. Although the findings are biologically plausible, the lack of adjustment for multiple testing (e.g., Bonferroni or false discovery rate) should be acknowledged, as it may increase the risk of false-positive associations. To strengthen statistical rigor, the authors may consider applying a correction for multiple genetic comparisons (e.g., false discovery rate) or, alternatively, explicitly state that the genetic analyses are exploratory in nature.

Reviewer #6: Study design and objective very clearly stated. Ethical approval is taken for the study. Conclusion derived from the study is acceptable.

Reviewer #7: -Are the objectives of the study clearly articulated with a clear testable hypothesis stated?

Yes

-Is the study design appropriate to address the stated objectives?

Yes

-Is the population clearly described and appropriate for the hypothesis being tested?

Yes

-Is the sample size sufficient to ensure adequate power to address the hypothesis being tested?

Yes

-Were correct statistical analysis used to support conclusions?

Yes

-Are there concerns about ethical or regulatory requirements being met?

No

**Results**

-Does the analysis presented match the analysis plan?

-Are the results clearly and completely presented?

-Are the figures (Tables, Images) of sufficient quality for clarity?

Reviewer #1: done

Reviewer #2: This is a good and thorough study / analysis, and excellent way of narrowing down on vit.D mechanisms in TUs, with adequate attention for confounders.

Reviewer #3: Findings clear and well-structured.

Reviewer #4: Yes, the results are presented very well and clearly.

Reviewer #5: - Results are clearly organized and logically presented, with appropriate reporting of effect estimates, confidence intervals, and p-values. The distinction between unadjusted and adjusted analyses is clear. Minor clarification could improve interpretation, particularly regarding the clinical relevance of wide confidence intervals in genetic analyses and the limitations of correlation analyses.

Reviewer #6: Yes; result presented on the basis of data analysis is well presented.

Reviewer #7: -Does the analysis presented match the analysis plan?

The paper would be improved if the authors instead of showing the value of Serum 25(OH)D levels (line 179), include in the manuscript a scatter dot plot showing the serum 25(OH)D levels. This would allow the reader to better understand which patients are driven the difference in the Leprosy patients with TU. In addition, it would be interesting to associate the vitamin D values with the clinical forms of leprosy. Is there any difference, among leprosy with TU group, in the 25(OH)D levels between the patients with BT/BB and BL/LL? More specifically, are the vitamin D levels in BL/LL patients lower than in BT/BB patients?

-Are the results clearly and completely presented?

Yes

-Are the figures (Tables, Images) of sufficient quality for clarity?

Yes

**Conclusions**

-Are the conclusions supported by the data presented?

-Are the limitations of analysis clearly described?

-Do the authors discuss how these data can be helpful to advance our understanding of the topic under study?

-Is public health relevance addressed?

Reviewer #1: done

Reviewer #2: Fully correct!

Reviewer #3: Reverse causality might be worth mentioning, while remains unaddressed by authors, there is still a possibility that chronic inflammatory state and physical limitations that lead to reduced sun exposure, dictated by ulcers, might lead to a secondary depletion of Vitamin D.

The data support the conclusion that low levels of vitamin D and Fokl polymorphism are independent factores associated with ulcers. The authors described the single-centre design and cross-sectional nature, very relevant aspect.

Bringing the discussion of the role of Vitamin D in leprosy cases may be highly relevant for endemic regions that requires a low-cost intervention that may reduce morbidity of trophic ulcers.

Reviewer #4: Yes, the conclusion is supported by the data presented.

Reviewer #5: - Key limitations related to the observational case–control design, potential reverse causality, and limited causal inference are acknowledged. However, additional limitations—particularly those related to genetic analyses (sample size, multiple comparisons, and population stratification)—could be more explicitly discussed to strengthen transparency.

Reviewer #6: Conclusion is clearly mentioned. Authors have suggested that will be helpful for understanding about tropic ulcer and further management , that may very important for patients suffering from ulcer

Reviewer #7: -Are the conclusions supported by the data presented?

Yes, except for the graph that shows the correlation between 25(OH)D levels and PUSH scores. Although there is a significance, the correlation is extremely weak, which was recognised by the authors, and even with p<0.05 the correlation is not biologically relevant. Thus, it is not possible for the authors to claim that "a modest inverse correlation was also found between vitamin D levels and ulcer severity" (lines 227-228). Also, if the authors perform this correlation analysis separating BT/BB and BL/LL, does the correlation becomes stronger?

-Are the limitations of analysis clearly described?

Yes

-Do the authors discuss how these data can be helpful to advance our understanding of the topic under study?

Yes

-Is public health relevance addressed?

Yes

**Editorial and Data Presentation Modifications?**

Reviewer #1: accept

Reviewer #2: Accept, you might want to convey to the authors the pdf with my comments, but these should not have effect in terms of any revisions. The manuscript is good to go as far as I am concerned.

Reviewer #3: Accept.

Reviewer #4: No, obvious comments for revisions. This can be accepted.

Reviewer #5: The text contains some minor typos such as “eritema arredores < 296 cm”, probably “<2 cm” and rarely used terms such as Data taking” for “Data collection" and other.

Reviewer #6: I recommend to Accept.

Reviewer #7: (No Response)

**Summary and General Comments**

Reviewer #1: done

Reviewer #2: No further comments, this is a well-conducted study!

Reviewer #3: The study is relevant and well-designed. Also has novel data regarding vitamin D and leprosy.

Mentioning that Sanger or NGS could not be performed due to cost is a great point, but also a weakness of the paper. It is still an acceptable choice on resource-limited settings.

Other relevant aspects includes the low cost of Vitamin D supplements as well as the possibility of using Fokl genotyping to identify patients and prioritize intensive care and nutritional support.

That may be the case in other countries that has high levels of leprosy and gives the rise of studies that focus on nutritional aspects of this disease.

Reviewer #4: the manuscript is written very well with clear study design and interpretation of results. It can be accepted.

Reviewer #5: - This study explores the association between vitamin D status, a functional VDR polymorphism, and trophic ulcers in patients with leprosy, focusing on a clinically relevant complication that has received limited attention.

Strengths of the study include the objective assessment of serum 25(OH)D levels, the use of multivariable logistic regression to account for key confounders, and the investigation of a biologically plausible genetic variant, with genotyping quality supported by Hardy–Weinberg equilibrium. The findings related to vitamin D are internally consistent and of potential clinical relevance.

The study is nonetheless subject to limitations inherent to its case–control design, including the possibility of residual confounding and limited causal inference. In addition, the sample size may restrict the robustness of the genetic analyses. The absence of formal correction for multiple genetic comparisons, along with the possibility of population stratification, suggests that the genetic findings should be interpreted with caution and viewed primarily as exploratory.

Overall, the manuscript contributes useful data to an underexplored area of leprosy research and, with appropriate acknowledgment of its limitations, represents a relevant addition to the neglected tropical diseases literature.

Reviewer #6: Overall all considerations regarding study is well stated.

Reviewer #7: The authors investigated a subject that it is relevant for the public health but usually neglected. Interesting results were obtained and the experimental design was well articulated with the questions proposed. The association between trophic ulcer (TU) and vitamin D levels, and also FokI polymorphism showed that vitamin D levels is likely linked with the formation of TU. However, the authors only showed the mean values of Serum 25(OH)D levels (line 179). It is preferable to build a plot (scattered plot as already suggested) to know whether the levels of 25(OH)D was homogeneous in leprosy patients with ulcers or if there is a subgroup among leprosy patients with TU that drives the reduction of 25(OH)D.

PLOS authors have the option to publish the peer review history of their article (what does this mean?). If published, this will include your full peer review and any attached files.). If published, this will include your full peer review and any attached files.). If published, this will include your full peer review and any attached files.). If published, this will include your full peer review and any attached files.

...

Reviewer #1: No

Reviewer #2: **Yes:**Erik PostErik PostErik PostErik Post

Reviewer #3: No

Reviewer #4: No

Reviewer #5: No

Reviewer #6: **Yes:**Dr Mahesh ShahDr Mahesh ShahDr Mahesh ShahDr Mahesh Shah

Reviewer #7: No

**Figure resubmission:** While revising your submission, we strongly recommend that you use PLOS’s NAAS tool (https://ngplosjournals.pagemajik.ai/artanalysis) to test your figure files. NAAS can convert your figure files to the TIFF file type and meet basic requirements (such as print size, resolution), or provide you with a report on issues that do not meet our requirements and that NAAS cannot fix.

**Reproducibility:** To enhance the reproducibility of your results, we recommend that authors of applicable studies deposit laboratory protocols in protocols.io, where a protocol can be assigned its own identifier (DOI) such that it can be cited independently in the future. Additionally, PLOS ONE offers an option to publish peer-reviewed clinical study protocols. Read more information on sharing protocols at https://plos.org/protocols?utm_medium=editorial-email&utm_source=authorletters&utm_campaign=protocols To enhance the reproducibility of your results, we recommend that authors of applicable studies deposit laboratory protocols in protocols.io, where a protocol can be assigned its own identifier (DOI) such that it can be cited independently in the future. Additionally, PLOS ONE offers an option to publish peer-reviewed clinical study protocols. Read more information on sharing protocols at https://plos.org/protocols?utm_medium=editorial-email&utm_source=authorletters&utm_campaign=protocols

---

## [Decision Letter · Decision Letter 1]

24 Mar 2026

Response to Reviewers'. This file does not need to include responses to any formatting updates and technical items listed in the 'Journal Requirements' section below.'. This file does not need to include responses to any formatting updates and technical items listed in the 'Journal Requirements' section below.* A marked-up copy of your manuscript that highlights changes made to the original version. You should upload this as a separate file labeled 'Revised Manuscript with Track Changes'.'.* An unmarked version of your revised paper without tracked changes. You should upload this as a separate file labeled 'Manuscript'.'.If you would like to make changes to your financial disclosure, competing interests statement, or data availability statement, please make these updates within the submission form at the time of resubmission. Guidelines for resubmitting your figure files are available below the reviewer comments at the end of this letter.We look forward to receiving your revised manuscript.Kind regards,

Mohammad Jokar, DVM

Guest Editor

Shaden Kamhawi

co-Editor-in-Chief

Paul Brindley

co-Editor-in-Chief

**Reviewers' comments:**Reviewer's Responses to Questions

**Key Review Criteria Required for Acceptance?**

**Methods**

-Are the objectives of the study clearly articulated with a clear testable hypothesis stated?

-Is the study design appropriate to address the stated objectives?

-Is the population clearly described and appropriate for the hypothesis being tested?

-Is the sample size sufficient to ensure adequate power to address the hypothesis being tested?

-Were correct statistical analysis used to support conclusions?

-Are there concerns about ethical or regulatory requirements being met?

Reviewer #5: (No Response)

Reviewer #7: The authors did what it was required from the last review - perform analysis of the subgroups of leprosy patients. They also included, as previously suggested, the figure showing the levels of serum 25(OH)D levels in TU and non-TU group (new Figure 1).

My final suggestion is to include in figure 3 the correlation graph for the BT/BB subgroup (Fig 3B), and for the BL/LL subgroup (Fig 3C) and highlight that the weak correlation found for leprosy patients with TU is not biologically relevant but if they look only BT/BB group then the correlation could be relevant. For some reason, in severe forms of leprosy this correlation is not significant. It is important also that they make a brief discussion about that. What is the authors hypothesis for no correlation in BL/LL individuals? After clarify those issues I aggree with the publication of the manuscript.

**Results**

-Does the analysis presented match the analysis plan?

-Are the results clearly and completely presented?

-Are the figures (Tables, Images) of sufficient quality for clarity?

Reviewer #5: (No Response)

Reviewer #7: (No Response)

**Conclusions**

-Are the conclusions supported by the data presented?

-Are the limitations of analysis clearly described?

-Do the authors discuss how these data can be helpful to advance our understanding of the topic under study?

-Is public health relevance addressed?

Reviewer #5: (No Response)

Reviewer #7: (No Response)

**Editorial and Data Presentation Modifications?**

Reviewer #5: (No Response)

Reviewer #7: (No Response)

**Summary and General Comments**

Reviewer #5: (No Response)

Reviewer #7: (No Response)

PLOS authors have the option to publish the peer review history of their article (what does this mean?). If published, this will include your full peer review and any attached files.). If published, this will include your full peer review and any attached files.). If published, this will include your full peer review and any attached files.). If published, this will include your full peer review and any attached files.

...

Reviewer #5: **Yes:**Silmara Navarro PenniniSilmara Navarro PenniniSilmara Navarro PenniniSilmara Navarro Pennini

Reviewer #7: No

**Figure resubmission:**While revising your submission, we strongly recommend that you use PLOS’s NAAS tool (https://ngplosjournals.pagemajik.ai/artanalysis) to test your figure files. NAAS can convert your figure files to the TIFF file type and meet basic requirements (such as print size, resolution), or provide you with a report on issues that do not meet our requirements and that NAAS cannot fix.

**Reproducibility:**To enhance the reproducibility of your results, we recommend that authors of applicable studies deposit laboratory protocols in protocols.io, where a protocol can be assigned its own identifier (DOI) such that it can be cited independently in the future. Additionally, PLOS ONE offers an option to publish peer-reviewed clinical study protocols. Read more information on sharing protocols at https://plos.org/protocols?utm_medium=editorial-email&utm_source=authorletters&utm_campaign=protocolsTo enhance the reproducibility of your results, we recommend that authors of applicable studies deposit laboratory protocols in protocols.io, where a protocol can be assigned its own identifier (DOI) such that it can be cited independently in the future. Additionally, PLOS ONE offers an option to publish peer-reviewed clinical study protocols. Read more information on sharing protocols at https://plos.org/protocols?utm_medium=editorial-email&utm_source=authorletters&utm_campaign=protocols

---

## [Decision Letter · Decision Letter 2]

29 Mar 2026

Dear Dr. dr. Sirait,

We are pleased to inform you that your manuscript 'Comparison of serum vitamin D level and vitamin D receptor gene FokI polymorphism in leprosy patients with and without trophic ulcers: A case-control study' has been provisionally accepted for publication in PLOS Neglected Tropical Diseases.

Best regards,

Mohammad Jokar, DVM

Guest Editor

Stuart Blacksell

Section Editor

Shaden Kamhawi

co-Editor-in-Chief

Paul Brindley

co-Editor-in-Chief

Reviewer's Responses to Questions

**Key Review Criteria Required for Acceptance?**

**Methods**

-Are the objectives of the study clearly articulated with a clear testable hypothesis stated?

-Is the study design appropriate to address the stated objectives?

-Is the population clearly described and appropriate for the hypothesis being tested?

-Is the sample size sufficient to ensure adequate power to address the hypothesis being tested?

-Were correct statistical analysis used to support conclusions?

-Are there concerns about ethical or regulatory requirements being met?

Reviewer #7: (No Response)

**Results**

-Does the analysis presented match the analysis plan?

-Are the results clearly and completely presented?

-Are the figures (Tables, Images) of sufficient quality for clarity?

Reviewer #7: (No Response)

**Conclusions**

-Are the conclusions supported by the data presented?

-Are the limitations of analysis clearly described?

-Do the authors discuss how these data can be helpful to advance our understanding of the topic under study?

-Is public health relevance addressed?

Reviewer #7: (No Response)

**Editorial and Data Presentation Modifications?**

Reviewer #7: (No Response)

**Summary and General Comments**

Reviewer #7: (No Response)

PLOS authors have the option to publish the peer review history of their article (what does this mean?). If published, this will include your full peer review and any attached files.). If published, this will include your full peer review and any attached files.). If published, this will include your full peer review and any attached files.). If published, this will include your full peer review and any attached files.

...

Reviewer #7: No

---

## [Editor Report · Acceptance letter]

Dear Dr. dr. Sirait,

We are delighted to inform you that your manuscript, "Comparison of serum vitamin D level and vitamin D receptor gene FokI polymorphism in leprosy patients with and without trophic ulcers: A case-control study," has been formally accepted for publication in PLOS Neglected Tropical Diseases.

Best regards,

Shaden Kamhawi

co-Editor-in-Chief

Paul Brindley

co-Editor-in-Chief
